Pathogen reduction co-benefits of nutrient best management practices

Richkus Jennifer 1 jrichkus@rti.org
Wainger Lisa A. 2
Barber Mary C. 1
1 RTI International , Washington, District of Columbia , United States
2 Center for Environmental Science, University of Maryland , Solomons, Maryland , United States
Hamre Kristin
Electronic publication date: 2016 Nov 22
Publication date: 2016
Volume: 4
Electronic Location ID: e2713
Received 2016 Jun 15; Accepted 2016 Oct 24
Copyright: © 2016 Richkus et al.
Copyright year: 2016
Copyright holder: Richkus et al.
License: This is an open access article distributed under the terms of the Creative Commons Attribution License, which permits unrestricted use, distribution, reproduction and adaptation in any medium and for any purpose provided that it is properly attributed. For attribution, the original author(s), title, publication source (PeerJ) and either DOI or URL of the article must be cited.
License URL: https://creativecommons.org/licenses/by/4.0/

Keywords: Pathogens, TMDL, Ecosystem services, Nutrients, Best management practices, Water quality, Chesapeake Bay, Fecal coliform

Funding: U.S. Environmental Protection Agency Office of Research and Development Funding for this work was provided by U.S. Environmental Protection Agency Office of Research and Development. The funders had no role in study design, data collection and analysis, decision to publish, or preparation of the manuscript.

==============================
Background

Many of the practices currently underway to reduce nitrogen, phosphorus, and sediment loads entering the Chesapeake Bay have also been observed to support reduction of disease-causing pathogen loadings. We quantify how implementation of these practices, proposed to meet the nutrient and sediment caps prescribed by the Total Maximum Daily Load (TMDL), could reduce pathogen loadings and provide public health co-benefits within the Chesapeake Bay system.

Methods

We used published data on the pathogen reduction potential of management practices and baseline fecal coliform loadings estimated as part of prior modeling to estimate the reduction in pathogen loadings to the mainstem Potomac River and Chesapeake Bay attributable to practices implemented as part of the TMDL. We then compare the estimates with the baseline loadings of fecal coliform loadings to estimate the total pathogen reduction potential of the TMDL.

Results

We estimate that the TMDL practices have the potential to decrease disease-causing pathogen loads from all point and non-point sources to the mainstem Potomac River and the entire Chesapeake Bay watershed by 19% and 27%, respectively. These numbers are likely to be underestimates due to data limitations that forced us to omit some practices from analysis.

Discussion

Based on known impairments and disease incidence rates, we conclude that efforts to reduce nutrients may create substantial health co-benefits by improving the safety of water-contact recreation and seafood consumption.

Introduction

Chesapeake Bay water quality has diminished over the past 60 years to the point that the system is less able to support abundant crabs and diverse fish, feed waterfowl, and produce safe recreational opportunities. To restore the bay, the U.S. Environmental Protection Agency (EPA) Chesapeake Bay Total Maximum Daily Load (TMDL) has been developed, which sets yearly caps on the levels of nitrogen, phosphorus, and sediment entering the system. Jurisdictions in the watershed (Delaware, Maryland, New York, Pennsylvania, Virginia, West Virginia and the District of Columbia) have created Watershed Implementation Plans (WIPs) to meet the requirements of the TMDL. The WIPs (Phase 2) include a diverse list of best management practices (BMPs) that impede the interaction and/or delivery of nutrients and sediment downstream, many of which also have the potential to reduce pathogens through this mechanism (Mallin et al., 2000; Knox et al., 2008).

Water that interacts with fecal matter can contain diverse pathogens such as Vibrio, E. coli (pathogenic), Shigella, Rotavirus, Yersinia, Cryptosporidium, and Giardia (Savichtcheva & Okabe, 2006) that have been linked to gastrointestinal illnesses, skin infections, fevers, and other human health concerns (Vann et al., 2002). Limited data are available on pathogen concentrations and exposure in the watersheds we studied; however, the level of concern for pathogens in the Chesapeake Bay watershed is evident from the actions that officials have taken to address them. Because of the potential for creating illness, government officials have responded to potential water contamination by closing beaches and waterways to recreators and closing shellfish beds to commercial and recreational harvest. For example, 176 Virginia shellfishing areas are indefinitely closed due to elevated fecal coliform (VDH, 2012; EPA, 2013c), and 77 shellfish beds are occasionally or permanently closed in Maryland (MDE, 2016). As recently as 2008, approximately 8% of all shellfish beds in Maryland and Virginia were estimated to be closed due the potential for pathogen-related illness. These areas represent a cumulative impact to an industry valued at approximately $13 million in Maryland and Virginia in 2008 (Pelton & Goldsborough, 2010).

This study utilizes the data available in the literature and simple estimations to demonstrate that measurable reductions to pathogen loadings in the Chesapeake Bay are likely, as a result of implementing the TMDL. These reductions, in turn, can provide benefits to recreators and fishers by increasing confidence in waters and potentially reducing closures, and to government officials by reducing the level of effort expended on pathogen-related actions.

Although more research will be needed to quantify and value the change, this study is intended to indicate the ancillary benefits that can be considered by decision makers when weighing the costs and benefits of environmental management programs such as TMDLs.

Materials and Methods

This analysis investigated the effects of the WIPs by examining the effectiveness of practices for reducing pathogen delivery to waterways. Information on current pathogen loadings in the Potomac River basin and pathogen reduction potential of BMPs was gathered to estimate the potential reduction in pathogens throughout the Potomac River basin and the Chesapeake Bay watershed (Fig. 1).

Figure 1 Map of the Chesapeake Bay watershed and Tributary basins.

Four steps were applied to estimate the potential pathogen reduction attributable to the TMDL, as described below.

Step 1: Define BMPs in Chesapeake Bay TMDL implementation

To define the types and extent of BMP implementation proposed to meet the TMDL, we used information generated by Chesapeake Bay Watershed Model scenarios developed for and run by the US EPA Chesapeake Bay Program (CBP). The scenarios are based on the Phase 2 WIPs provided by all of the Chesapeake Bay jurisdictions to EPA and indicate acreage or number of BMPs that are anticipated to be implemented within given areas of the watershed. Using the information provided by US EPA CBP, we calculated the difference in estimates between the “2009 baseline” and the “with TMDL” model run scenarios. Thus, our estimates of BMP implementation represent the change in pathogen loads resulting only from actions taken to meet the TMDL.

Step 2: Identify pathogen reduction efficiencies for BMPs

We then conducted a literature review of the BMPs identified in the Chesapeake Bay WIPs to investigate the potential efficiency of agricultural, urban, and septic BMPs in reducing pathogens at edge of field or edge of (small) streams. The WIP efforts identified to have the potential to affect the delivery and concentration of these waterborne pathogens include (categorized by source): Agricultural: pasture and grazing management, nutrient management on crop fields, livestock waste management, restricted stream access, plantings and other structural practices to reduce nutrient and sediment runoff.

Urban: detention and retention ponds, impervious surface reduction, street sweeping, forested riparian buffers, bioswales, afforestation.

Septic: connecting septic systems to sewers, septic pumping, and on-site septic upgrades.

Wastewater treatment plants: new and enhanced treatment of municipal waste.

The literature review was conducted by employing each of the BMPs and pathogen-related terms as keywords (e.g., “restricted stream access” or “riparian buffer” and “fecal coliform” or “E. coli” or “bacteria”) using Google Scholar, EBSCO, and Google as search engines. Pathogen reduction estimates from peer-reviewed journal articles, documentation prepared by state agencies for compliance with TMDLs, and best practice guidance reports from state agencies and universities were all considered for the purposes of this paper. Data were included in the analysis only if they could be matched to the BMP form and function and were relevant to the pathogens being evaluated.

The range of pathogen removal efficiencies varied widely from −6% to 99%, where negative efficiencies increased pathogen concentrations as a result of the BMP implementation (Table 1). It was observed that fecal indicator bacteria (FIB), which includes fecal coliform and E. coli, were most often evaluated as a surrogate for a variety of pathogens (Marion et al., 2010; EPA, 2012a). Therefore, to make the best use of the literature, we averaged the efficiencies of fecal coliform and E. coli into a single FIB value to associate with each BMP in the analysis.

Table 1 Literature review of pathogen reduction efficiencies for crop, pasture, urban, and septic BMPs.

Best management practice1	Loading reduction efficiency (%)	Average fecal coliform and E. Coli (FIB)2 efficiency (%)	Reference	
Crop practices	
Forest buffers	Fecal coliform: 43–57	50	Virginia Department of Environmental Quality (2003)	
Grass buffers	E. coli: 58–99
Fecal coliform: 28–100	71	Minnesota Pollution Control Agency (2009) and Peterson et al. (2012b)	
Land retirement	90–93	92	Virginia Department of Environmental Quality (2003) and Peterson et al. (2012b)	
Water control structures	Detention structures: 67	67	Leisenring, Clary & Hobson (2012)	
Wetland restoration	E. coli: 40
Fecal coliform: 30	35	Virginia Department of Environmental Quality (2003)	
Non-urban stream reduction	No estimate	Not included		
Pasture practices	
Barnyard runoff control	Fecal coliform: 81	81	U.S. Geological Survey (1998)	
Forest buffers	Fecal coliform: 43–57	50	Virginia Department of Environmental Quality (2003)	
Grass buffers	E. coli: 58–99
Fecal coliform: 28–100	71	Minnesota Pollution Control Agency (2009) and Peterson et al. (2012b)	
Horse pasture management	E. coli: 72	72	Peterson et al. (2012a)	
Loafing lot management	Fecal coliform: 50	50	Virginia Department of Environmental Quality (2003)	
Pasture alternative watering	E. coli: 85–95
Fecal coliform: 51–94	82	Sheffield et al. (1997) and Byers et al. (2005)	
Precision intensive rotational grazing	Fecal coliform: 90	90	Minnesota Pollution Control Agency (2009)	
Prescribed grazing	E. coli: 66–72
Fecal coliform: 90–96	80	Peterson, Redmon & McFarland (2011b)	
Stream access control with fencing	E. coli: 37–46
Fecal coliform: 30–94	52	Schaetzle (2005) and Peterson, Redmon & McFarland (2011a)	
Ammonia emission reductions	No estimate	Not included		
Conservation tillage with continuous no till	No estimate: heavily dependent on if and when animal manure has been applied	Not included	Ramirez et al. (2009)	
Dairy precision feeding	No estimate	Not included		
Livestock mortality composting	No estimate	Not included		
Livestock waste management systems	E. coli: 97–99
Fecal coliform: 44–99	Not included	Virginia Department of Environmental Quality (2003) and Redmon, Wagner & Peterson (2012)	
Manure transport inside CBWS	No estimate	Not included		
Manure transport outside CBWS	Assumed to be 99	Not included		
Non-urban stream restoration	Fecal coliform: 30	Not included	Virginia Department of Environmental Quality (2003)	
Poultry phytase	No estimate	Not included		
Poultry waste management systems	Fecal coliform: 75 E. coli: 96	Not included	Virginia Department of Environmental Quality (2003) and Redmon, Wagner & Peterson (2012)	
Urban practices	
BioRetention	E. coli: 71	71	Leisenring, Clary & Hobson (2012)	
Bioswale	Fecal coliform: −53
E. coli: −6	−6	Leisenring, Clary & Hobson (2012)	
Dry ponds	Fecal coliform: 80	80	Tilman, Plevan & Conrad (2011)	
Erosion and sediment control	Assumed average of all urban stormwater practices:
Fecal coliform: 53
E. coli: 60	57		
Filtering practices	Fecal coliform: 60
E. coli: 99	80	Clary et al. (2008)	
Forest buffers	Fecal coliform: 43–57	50	Virginia Department of Environmental Quality (2003)	
Impervious surface reduction	Assumed average of all urban stormwater practices:
Fecal coliform: 53
E. coli: 60	57		
Infiltration practices	Assumed to be equivalent to Leisenring, Clary & Hobson (2012) retention ponds:
E. coli: 95 Fecal coliform: 65	80	Leisenring, Clary & Hobson (2012)	
Retrofit Stormwater management	Assumed average of all urban stormwater practices:
Fecal coliform: 53
E. coli: 60	57		
Wet ponds & wetlands	Fecal coliform: 53
E. coli: 43–68	48	Leisenring, Clary & Hobson (2012) and Knox et al. (2008)	
Abandoned mine reclamation	No estimate	Not included		
Street sweeping	Fecal coliform: 1.4–4.3	Not included	Zarriello, Breault & Weiskel (2003)	
Tree planting	No estimate	Not included		
Urban stream restoration	No estimate	Not included		
Septic practices	
Combined sewer overflow elimination	Fecal coliform: 99	Not included	City of Grand Rapids (2011)	
Septic connections	Fecal coliform: 99	Not included	Vann et al. (2002) and Petersen, Rifai & Stein (2009)	
Septic denitrification	No estimate obtained	Not included		
Septic pumping	Fecal coliform: 5	Not included	Virginia Department of Environmental Quality (2003)	
Treatment plant upgrades	No estimate: heavily dependent on type of upgrade and technology implemented	Not included		
Notes:

1 No comprehensive set of definitions of the BMPs used in the WIPs was available; however, definitions for these agricultural practices can be found here: http://mda.maryland.gov/resource_conservation/WIPCountyDocs/bmpdef_pg.pdf. Summaries of the types of practices used in the urban BMPs can be found here: http://www.dnrec.delaware.gov/swc/wa/Documents/ChesapeakePhaseIIWIP/Final_Phase2_CBWIP_03302012A.pdf.

2 FIB, or fecal indicator bacteria, reduction efficiency is represented by the average reduction efficiencies of E. coli and fecal coliform for the purposes of this analysis.

3 Negative removal efficiencies indicate that the concentrations of pathogens increased as a result of the BMP implementation.

Agricultural practices showed a range of efficiencies at removing fecal coliform and E. coli (28–100%), but the average performance per practice was above 50% for all practices except wetland and stream restoration. Studies also showed high efficiency of grassed buffers at removing cryptosporidium (93–99%). Stormwater practices showed a wider range of removal efficiencies (−6–99%) than agricultural practices when looking across the range of practices. However, a few practices were responsible for the cases of low performance (bioswales, street sweeping, and septic pumping). The majority of practices had average efficiencies of 48% or greater.

Step 3: Estimate baseline pathogen loads

To estimate the baseline pathogen load, we required an understanding of pathogen sources and deliveries to water bodies, for the given level of management practices implemented in the baseline scenario. A study of the Upper Potomac River Basin, the portion that lies above the fall line1, provided the best available information about how pathogens were being produced, transformed, intercepted and, finally, delivered downstream (Vann et al., 2002). That study estimated average annual edge-of-stream (EOS) pathogen loadings for a period that roughly corresponded to 2000–2010. The 2010 scenario was a projection of land use and population changes expected to occur by 2010 combined with 2000 estimates of non-point source BMPs and wastewater loads, and 2010 estimates for septic conditions. We use the 2010 model results as if they occurred in conditions equivalent to the baseline scenario developed by the CBP.

Vann et al. (2002) estimated loadings by land use type, using models similar to those of the CBP but modified to include pathogen movement and transformation and a wide variety of data sources on fecal sources. Data on livestock, geese, deer and human populations; National Pollution Discharge Elimination System (NPDES) and wastewater emissions; and other sources were used to inform modeling of pathogen loads by land use type. The CBP watershed model was adapted to include bacterial fate and transport, and the loads by land use were calibrated using pathogen concentrations measured at monitoring stations, primarily within the main channel of this major Chesapeake tributary. The model was also combined with data from surface water intakes to estimate the downstream delivery factors for the Potomac River basin.

To use the Vann et al. (2002) results to estimate baseline loads for the land uses in the entire Potomac River basin and the Chesapeake Bay watershed, we converted the EOS loads to per acre loadings per land use type (Table 2). We evaluated only the three land uses being modified by the BMPs used in the analysis for the purposes of converting EOS loads, as described below. We then multiplied the per acre loads for acreages of pasture, cropland, and urban for the baseline scenario to estimate baseline loads.

Table 2 Modeled loadings per land use source in the upper Potomac River basin.

Loading type/land use	Edge-of-stream delivery of fecal coliform (cfu/yr)1	Edge-of-stream delivery per acre (cfu/ac/yr)	Edge-of-stream loading delivered downstream (%)2	
Cropland	6.0E + 16	5.18E + 10	25	
Pasture	3.2E + 17	3.88E + 113	28	
Feedlots	6.3E + 16	3.88E + 113	24	
Cattle4	1.0E + 16		21	
Urban	2.2E + 16	1.82E + 10	27	
Notes:

All data derived from Vann et al. (2002).

1 Pathogens were measured as fecal coliform in colony forming units per year (cfu/year).

2 Proportion delivered downstream was calculated with mass balance equations, based on data provided by Vann et al. (2002).

3 Land uses were combined for the delivery estimates per acre because acreages were not reported separately for these land uses.

4 Cattle land use is an estimate of deposition of feces directly into water bodies.

This method relies on transferring results of sophisticated models for the upper Potomac to two different scales of analysis (Potomac River and entire Chesapeake Bay watersheds), to provide a rough estimate of TMDL implementation effects at these scales. Clearly, using data from a portion of the Potomac River basin to represent either the whole Potomac River basin or the entire Chesapeake Bay watershed requires making considerable assumptions about the similarity of patterns and processes at these two scales. We have greater confidence in the Potomac River basin results because the Potomac River basin would be expected to be more similar to the originally modeled area than the Chesapeake Bay as a whole. The Potomac River basin may be a reasonable model for the entire Chesapeake Bay watershed because it makes up over one-fifth of the Chesapeake Bay watershed and has proportions and distribution of land use types that are similar to the entire Chesapeake Bay watershed. However, the Potomac River basin differs from the Chesapeake Bay watershed in that it has slightly more urban land and pasture and less forest (Table 3), and BMPs were applied in different proportions to the whole Bay, as shown in the Results and Discussion section.

Table 3 Land use composition of Potomac River basin and the Chesapeake Bay watershed.

Land use	Potomac River basin (acres)	Potomac River basin land use (%)	Chesapeake Bay land use (acres)	Chesapeake Bay basin land use (%)	
Forest	5,189,905	59	26,512,720	65	
Cropland	1,405,191	16	6,640,633	16	
Pasture	920,935	10	2,438,478	6	
Urban	1,245,535	14	4,853,216	12	
Other	99,827	1	653,219	2	
Total	8,861,392	100 (22% of Chesapeake Bay watershed)	41,098,267	100	
Note:

Jeff Sweeney of the US EPA Chesapeake Bay Program; 2009 baseline scenario data.

Step 4: Estimate change in pathogens due to the TMDL

As described earlier, the acreage of BMPs implemented due to the TMDL was derived by subtracting the baseline BMP implementation from the “with TMDL” scenario. Each BMP was associated with a particular land use and quantified in terms of the acres of that land use that were affected by implementation of the BMP. For example, prescribed grazing was associated with pasture, and the percentage of total pasture under prescribed grazing was used to estimate changes in pathogen loads.

Because of data limitations, only a subset of BMPs that are capable of reducing pathogen loads were used in our analysis. BMPs were omitted from analysis if they were not measured in terms of acreage in the state WIPs, or if efficiencies were specific to baseline conditions that could not be accurately measured. For example, omitted BMPs include those measured as pounds of manure transported outside of the watershed and miles of stream restored. Also, some cropland practices in widespread use, such as continuous no-till, can be effective at reducing pathogens, but only when applied to cropland receiving manure; lack of sufficient data on manure handling prevented their inclusion. Omitting these practices, as well as point source practices, such as septic and wastewater treatment plant upgrades, tends to make our study more conservative in terms of the TMDL effectiveness for reducing pathogens because practices that are expected to be implemented as part of the WIPs were not counted, and some of these practices have been demonstrated to be highly effective at reducing pathogen loads (Table 1).

To estimate the change in pathogen loads (measured as FIB) delivered to the main channel as a result of applying a subset of BMPs from the WIPs, we applied Eq. (1): (1) ΔFIBDS=∑​l(∑​b(BMP Acres)b,l(Total land area)l(%FIB reduction)b)(EOS load)l(%DS Delivery)l

where b is the BMP applied and

l is the land use type.

Equation (1) shows that delivery of pathogens to the main channel depends on edge-of-stream (EOS) loads and downstream (DS) attenuation of pathogen loads.

BMP Acres represents the acres of a given land use treated with a given BMP. The Total land area per land use (l) was derived from the baseline scenario. The %FIB reduction was the average removal efficiency for fecal coliform and E. coli for a given BMP. The proportion of treated acres to total acres in a given land use was multiplied by the percentage reduction for a given practice, and then these values were summed for all BMPs affecting a land use to generate a weighted sum representing the percentage reduction in pathogen loads expected for a given land use. The expected percent reduction for a given land use was multiplied by the baseline load for that land use to generate the EOS load (cfu/yr). Finally, the DS load was estimated by multiplying the EOS load by 21%, which was the average delivery ratio for all Potomac River segments modeled in the study by Vann et al. (2002).

Results and Discussion

Using the subset of BMPs that we were able to include, we estimated that the downstream pathogen reductions in the Potomac River basin due to WIP implementation would be on the order of 19% and in the Chesapeake Bay on the order of 27% total reduction. In the Potomac, 23% of reductions were derived from pasture loads, 6% from cropland loads, and 7% from urban loads (excluding point source loads) (Table 4). The 19% pathogen load reduction (downstream delivery) is estimated relative to total loads from all point and non-point sources to the mainstem Potomac River basin including domestic and wild animal sources. Urban load reduction results are sensitive to assumptions that practices will be maintained.

Table 4 Total loading reduction estimates for the Potomac River basin and Chesapeake Bay watershed1.

	Pasture practices reduction (pasture + feedlots)	Crop practices reduction	Urban practices reduction	Total (all sources)†	
Potomac River basin	
Acres of BMPs	273,423	136,341	114,676	524,440	
Potential reduction main channel (cfu/yr)	1.73E + 16	9.80E + 14	3.44E + 14	1.86E + 16	
Sector loadings reduced (%)	23%	6%	7%	19%	
Chesapeake Bay watershed	
Acres of BMPs	1,098,666	820,429	1,071,777	2,990,872	
Potential reduction main channel (cfu/yr)	7.28E + 16	6.25E + 15	3.21E + 15	8.22E + 16	
Sector loadings reduced (%)	36%	8%	17%	27%	
Note:

1 The percentage of total load reduction is calculated as the expected reduction in load from agriculture and urban non-point source sectors divided by estimated pathogen loads from all watershed sources (including wildlife and point sources). Therefore, the total in the rightmost column is smaller than the weighted sum of the percentage reductions from the three individual source sectors shown in the other columns.

† All estimated sources. Additional sources may exist that have not been considered in this analysis.

Percentage reductions are higher for the entire Chesapeake Bay watershed due to the increased agricultural land use composition, where the most potential for pathogen reduction was identified and able to be included in the analysis. The 27% total reduction in pathogen loads to the Bay tidal waters were from 36% reduction from pasture loads, 8% from cropland loads, and 17% of urban loads (excluding point source loads) (Table 4). However, we expect these numbers to be underestimates of the mainstem effects because the analysis does not include effects of septic upgrades, combined sewer overflows eliminations and some BMPs that were omitted but that are known to have high efficiency at removing pathogens.

The exclusion of BMPs, such as the waste management systems and septic connections, are a source of underestimation. For example, analyses developed for bacterial TMDLs in Virginia (Virginia Department of Environmental Quality, 2011) estimated that the elimination of emissions from 46 failing septic systems in Sugarland Run would reduce E. coli instream loadings by 8.89 × 1011 cfu/yr, which is an estimated per unit loading of 1.93 × 1010 cfu/yr. If, based on the literature review, we assume 1.93 × 1010 cfu/yr loadings per failing septic2, and if the number of septic system connections identified in the TMDL were implemented, loadings could be reduced by 4.22 × 1015, which is 19% of the fecal coliform loadings from other urban non-point sources in the Potomac River basin or 1% of total loadings from all natural and anthropogenic sources.

The estimated reductions in loads would be a substantial fraction of total loads to either the Potomac River or Chesapeake Bay watersheds. However, percentage reductions could be much higher in small water bodies. Because pathogen loads tend to become concentrated in localized areas, these reductions could be significant in terms of improving local water safety and preventing beach or shellfish closures, if practices were implemented at sufficient levels within small basins.

Potential magnitude of benefits

Whether reductions in pathogens reduce human illness from water contact or shellfish ingestion is a function of the ability of the pathogens to produce disease in people, probability of exposure to the pathogens, pathogen concentration, the number of people exposed, and the characteristics of the people that influence their susceptibility to disease (e.g., Soller et al., 2003). A quantification of all these effects was beyond the scope of this project. However, we gathered some existing information and data to suggest the potential order of magnitude of benefits.

FIB are correlated with a number of illnesses caused by bacteria and viruses, and the illness that has been most consistently and clearly linked to water contact is increased risk of gastroenteritis (Kay et al., 1994; Fleisher et al., 1998; Wade et al., 2010), although other diseases have also been observed, including respiratory illnesses, ear infections, and skin rashes (Fleisher et al., 1998; Fleisher et al., 2010). Skin diseases (infections and rashes) have been most closely linked to non-point sources of pathogens (Fleisher et al., 2010), whereas gastroenteritis is more clearly linked to sewage (Wade et al., 2010). The gastrointestinal illnesses caused by shellfish consumption have been linked to concentrations of Vibrio spp. (Hlady & Klontz, 1996), but Vibrio concentrations are widespread in the marine environment and are not highly correlated with fecal coliform (DePaola et al., 2000) and only weakly correlated with nitrogen concentrations (Pfeffer, Hite & Oliver, 2003; Eiler, Johansson & Bertilsson, 2006; Johnson et al., 2010). However, concentrations of Vibrio spp. have been linked to increased sediment suspension in some cases (Vanoy, Tamplin & Schwarz, 1992; Pfeffer, Hite & Oliver, 2003; Fries, Characklis & Noble, 2008).

The number of people harmed is indicated by the cases of reported illness that can be linked to waterborne pathogens. Table 5 presents reported illnesses from Maryland and Virginia. These states were included because these are the states with Chesapeake Bay shoreline, but other swimmable water bodies would also be affected by pathogens.

Table 5 Reported diseases due to pathogens in water bodies in Maryland and Virginia (2004–2013)1.

Waterborne disease	Maryland average	Virginia average	
Cryptosporidiosis	48	101	
Giardiasis	261	137	
Listeriosis	17	455	
Shiga: toxin producing E. coli (STEC)	96	19	
Shigellosis	164	159	
Vibriosis	35	32	
Total	621	902	
Notes:

1 Totals include illnesses due to treated (e.g., pools) and untreated (e.g., estuaries) water bodies, although the majority of these illnesses are likely from treated water bodies, which would not be affected by BMP implementation.

Source:

VDH (2014) and Maryland Department of Health and Mental Hygiene (2013).

These data on illnesses suggest a potential order of magnitude of illnesses caused by pathogens, but are not an accurate accounting, for three reasons. First, these numbers may be underestimates of true disease incidence due to water contact because only a fraction of illnesses are likely to have been identified and reported. Many more cases of gastrointestinal illnesses are likely to occur than to be reported (Hlavsa et al., 2014). Further, anecdotal information suggests that skin rashes and infections due to water contact are not an uncommon ailment in the Chesapeake Bay (Kobell, 2011; Kobell, 2013), particularly in the warmest months. These cases are not usually reported but have been documented elsewhere (Wade et al., 2010). Second, this particular set of cases may not be representative of risk associated with swimming in the Chesapeake Bay or its tributaries. Data collected by the Centers for Disease Control and Prevention using different reporting criteria found that 70% of reported illnesses due to waterborne pathogens were from pools or other treated water and 30% were from open (untreated) water, such as lakes and oceans (Hlavsa et al., 2014). Third, the relationship between animal-derived pathogens and human illnesses is poorly understood. Many dose-response relationships are based on pathogens from sewage sources, not agricultural sources.

Because of these data limitations, an estimation of the reduction in cases of disease was beyond the scope of this effort. However, we know that at least some water bodies contain dangerous levels of pathogen concentrations (evidence provided by TMDLs and cases of disease) that will be reduced in different proportions depending on the extent of BMP implementation in the watershed. The ability to reduce areas of high pathogen concentrations in areas with a high probability of exposure will have the most potential to create benefits.

Potential value of reduction in pathogens

This analysis suggests that these reductions can provide the following benefits to people in the Chesapeake Bay watershed. First, those who are in contact with the water (commercial fishermen, recreational anglers, boaters, and swimmers) are likely to have improved welfare due to illnesses avoided and may increase the number of trips they take. Second, more risk-averse recreators, who might currently avoid the water, might be induced to recreate in the Chesapeake Bay, in response to improved water safety (Lipton, 2004). Third, increased safety of shellfish could benefit commercial watermen, the burgeoning aquaculture industry, and seafood consumers. We would expect welfare increases from additional recreation trips, increased safety per trip, lowered costs of production for producers, and safer shellfish for consumers.

Table 6 summarizes the potential values of avoiding illnesses and beach closures as identified in the economic literature.

Table 6 Summary of economic values identified in the literature.

Value pathway (per person)	Economic value estimate	Reference	
Willingness to pay to avoid illness	$20.70–$64.43	Machado & Mourato (2002)	
Loss of beach trips	$2.51–$19.71	McConnell & Tseng (1999)	
Value of beach closure	$4.35–$7.96	McConnell & Tseng (1999)	
Value of beach closure	$0.00–$24.46	Parsons, Massey & Tomasi (1999)	
Loss of beach trips	$40.02	Murray, Sohngen & Pendleton (2001)	

Given the potential number of beach users in the Chesapeake Bay3, the total economic value of pathogen reductions due to the TMDL could be substantial when aggregated over the total number of beach users. Virginia had 29 days of beach actions (notifications and closure days) out of a total of 6,900 beach days in 2012 (open days multiplied by number of beaches), and an average of 56 beach actions per year from 2007–2012 (VDH, 2014; EPA, 2012b; EPA, 2013b). Maryland had 139 days of beach actions out of a total 6,501 beach days in 2012, and an average of 196 beach actions per year from 2007–2012 (EPA, 2013a).

Furthermore, these estimates do not include avoided costs due to reduced shellfish bed closures, lost wages and medical bills due to illness, or costs associated with stream miles impaired due to pathogens. Over 9,000 stream miles in Virginia (EPA, 2013c), over 4,000 miles in Maryland (MDE, 2012), and 190 miles in Pennsylvania (EPA, 2013c) are impaired for E. coli and fecal coliform4. A reduction in impaired stream miles would decrease both administrative costs associated with listing impaired waters and costs associated with developing and implementing TMDLs as required by Clean Water Act Section 303(d) (EPA, 2015).

Future efforts to estimate the total economic value of the TMDL with respect to pathogen reductions are required to fully understand the potential of the action.

Conclusions

Our literature review revealed that many BMPs being installed to reduce nutrients are effective at reducing pathogens. We provide a rough estimate of a 19% reduction in loads to tidal waters of the Potomac River and a 27% reduction in loads to tidal waters of the Chesapeake Bay. Substantial new modeling and data collection would be required to improve this estimate and relate it to reduced cases of illness, beach closures, or shellfish bed closures. If we take the simple approach of assuming that adverse events decline at the same rate as pathogen concentrations in the Chesapeake Bay (27%), we estimate that this would translate to hundreds of fewer cases of reported illness and substantial welfare effects. The health benefits of a 27% reduction appear modest based on overall reported numbers of illnesses. However, benefits could be significant for a local area if BMPs were concentrated in a watershed with a combination of moderate pathogen concentrations and a resource heavily used for recreation or shellfishing.

This study also suggests that there are additional benefits of the TMDL that have not been fully investigated and are not currently valued. Additional investigations are critical to helping decision makers understand the full suite of benefits that may be realized through the implementation of the TMDL as well as other water management and 303(d) actions.

Supplemental Information

Supplemental Information 1 Total Loading Reduction Estimates for the Potomac River Basin.

Click here for additional data file.

Supplemental Information 2 Total Loading Reduction Estimates for the Chesapeake Bay Watershed.

Click here for additional data file.

The authors thank Jeff Sweeney, Brenda Rashleigh, Naomi Detenbeck, George Van Houtven and Ross Loomis for helpful data, discussions and comments in the development of this paper.

Additional Information and Declarations

Competing Interests

Author Contributions

Data Deposition

1 The fall line in the Chesapeake Bay watershed is a geomorphic feature marked by a steep drop in elevation that occurs where the Piedmont and Coastal Plain geophysical provinces meet. It roughly corresponds to the division between non-tidal waters (above) and tidal waters (below).

2 Several estimates of fecal coliform loadings per failing septic units were identified within the Chesapeake Bay watershed during the literature review. The range per unit was 4.47 × 109 to 6.39 × 1012 cfu/yr. The median range was selected for this estimate because it was based on Hydrological Simulation Program—Fortran (HSPF) modeling of instream loadings rather than per capita fecal coliform production rate (Virginia Department of Environmental Quality, 2003; Virginia Department of Environmental Quality, 2011; West Virginia Department of Environmental Protection, 2012).

3 The number of people swimming in Chesapeake Bay was not readily available but a survey estimates that 42% of US residents engage in swimming in lakes, ponds, oceans, or rivers in a given year (Cordell et al., 2005).

4 The watershed also includes the District of Columbia and portions of West Virginia, Delaware, and New York.

The authors declare that they have no competing interests. Dr. Mary Barber and Jennifer Richkus are employees of RTI International.

Jennifer Richkus analyzed the data, wrote the paper, prepared figures and/or tables.

Lisa A. Wainger analyzed the data, wrote the paper.

Mary C. Barber reviewed drafts of the paper.

The following information was supplied regarding data availability:

Wainger L, Richkus J, Barber M. 2015. Additional beneficial outcomes of implementing the Chesapeake Bay TMDL: quantification and description of ecosystem services not monetized. Washington, D.C.: U.S. Environmental Protection Agency. EPA/600/R-15/052. Available at https://cfpub.epa.gov/si/si_public_record_report.cfm?dirEntryId=308098/.

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
