# Peer review of "Pathogen reduction co-benefits of nutrient best management practices"

_PeerJ, doi:10.7717/peerj.2713_

## Round 0.1 · original submission · Minor Revisions

I think the very useful exercise described in your ms deserves publication, but some revisions are needed. Please respond to the comments from reviewer 1.

The Materials and Methods section is difficult to follow. A map of the area and a more precise description of the central points in your calculations would be helpful, perhaps given as an overview figure. Also, the frequent use of abbreviations makes the reading difficult, for example; you could use full names in the references now given as abbreviations and limit other abbreviation uses.

Reviewer 1 ·

Basic reporting

The manuscript prepared by Richkus et al. provides some very interesting results on a topic that truly deserves more attention. It is of great importance that the measures implemented to restore polluted water bodies are critically evaluated. Although preliminary (as noted by the authors), this work deserves publication. It will have some impact in the field of ecology, health, recreational and waste/restoration practices of coastal/inland water bodies.

However, there is a need for very few major and some minor revision prior to publication in PeerJ.

There is too little focus on benefits other than health. Need more on eg. benefits for fisheries/aquaculture etc. Eg. L 340 what about societal benefits by encouraging more (local) seafood intake that may lead to reduced lifestyle diseases , and also encourage local residents to make use of the area for hiking/fishing/hunting trips? (obesity is a problem in USA). How many more people could be elaborated in fishing/aquaculture industry? Could this area potentially sustain a proportion of the US seafood requirements? Can import be decreased? Would it be more sustainable than the current situation?

L343 states that Table 6 "summarizes the potential values of avoiding illnesses and beach closures...", this is however only the commercial values and barely so. Please elaborate on Table 6 and tell what you expect to be potential values other than what can be counted in dollars.

The important question remains: What management practices would you advice, and why? Be bold (it improves impact...).

Experimental design

L95: The authors refer to some "model run scenarios". However, this model(s) is not well explained. Please clarify the modeling approach and state all model(s) used mathematically.

I would appreciate a map over the area indicating Chesapeake Bay, Potomac River and the other geographical names mentioned in the manuscript.

Validity of the findings

No comments

Additional comments

The "Methods" paragraph in the Abstract consists of one single sentence!! Please edit to facilitate the readability

L46: Delete Knox et al. 2007 (you probably mean 2008, Knox 2007 is not in the reference list)

L65: Fishermen? (not man)

L82, 85, and throughout (including tables). If you want to write for example "as derived by Vann et al. (2002)" you should use the "exclude author" function in EndNote or RefMan (or edit manually).

L92: Delete "the" before "all"

I am not sure how PeerJ handles footnotes, please consult the editor

Reviewer 2 ·

Basic reporting

The paper is well written and structured. Tables are Clear and readable.

Experimental design

no comment

Validity of the findings

This paper represents a useful contribution to a scientifically based evaluation of the benesits and effects of reducing the pathogen load from land runoffs into marine Coastal ares. Although focused on local effects, the paper may be used in related and general processes related to the effects of pollution and the improvement of waterbodies.

Additional comments

No coments

---

## Round 0.2 · accepted · Accept

The map and the improved Materials and Methods section have clarified the presentation of your work.